# Modelling the Effect of Salt Concentration on the Fate of *Listeria monocytogenes* Isolated from Costa Rican Fresh Cheeses

**DOI:** 10.3390/foods10081722

**Published:** 2021-07-26

**Authors:** Guiomar D. Posada-Izquierdo, Beatriz Mazón-Villegas, Mauricio Redondo-Solano, Alejandra Huete-Soto, Diana Víquez-Barrantes, Antonio Valero, Paula Fallas-Jiménez, Rosa María García-Gimeno

**Affiliations:** 1Departamento de Bromatología y Tecnología de los Alimentos, Facultad de Veterinaria, Campus de Excelencia Internacional Agroalimentario (ceiA3), Universidad de Córdoba, Ctra. Madrid-Cádiz km 396A, 14014 Córdoba, Spain; bt2vadia@uco.es (A.V.); bt1gagir@uco.es (R.M.G.-G.); 2 Escuela de Ingeniería de Biosistemas, Cuidad Universitaria Rodrigo Facio, Universidad de Costa Rica, 11501-2060 San José, Costa Rica; beatriz.mazon@ucr.ac.cr; 3Research Center for Tropical Diseases (CIET) and Food Microbiology Research and Training Laboratory (LIMA), Faculty of Microbiology, Ciudad Universitaria Rodrigo Facio, University of Costa Rica, 11501-2060 San José, Costa Rica; mauricio.redondosolano@ucr.ac.cr (M.R.-S.); alejandra.huete@ucr.ac.cr (A.H.-S.); paufallas97@gmail.com (P.F.-J.); 4National Center for Food Science and Technology (CITA), Ciudad Universitaria Rodrigo Facio, University of Costa Rica, 11501-2060 San José, Costa Rica; diana.viquezbarrantes@ucr.ac.cr

**Keywords:** artisanal fresh cheese, predictive microbiology, microHibro, *Listeria monocytogenes*, salt concentration, validation, isolated strains

## Abstract

“Turrialba cheese” is a Costa Rican fresh cheese highly appreciated due to its sensory characteristics and artisanal production. As a ready-to-eat dairy product, its formulation could support *Listeria monocytogenes* growth. *L. monocytogenes* was isolated from 14.06% of the samples and the pathogen was able to grow under all tested conditions. Due to the increasing demand for low-salt products, the objective of this study was to determine the effect of salt concentration on the growth of pathogen isolates obtained from local cheese. Products from retail outlets in Costa Rica were analyzed for *L. monocytogenes*. These isolates were used to determine growth at 4 °C for different salt concentration (0.5–5.2%). Kinetic curves were built and primary and secondary models developed. Finally, a validation study was performed using literature data. The R^2^ and Standard Error of fit of primary models were ranked from 0.964–0.993, and 0.197–0.443, respectively. An inverse relationship was observed between growth rate and salt concentration. A secondary model was obtained, with R^2^ = 0.962. The model was validated, and all values were *B_f_* > 1, thus providing fail-safe estimations. These data were added to the free and easy-to-use predictive microbiology software “microHibro” which is used by food producers and regulators to assist in decision-making.

## 1. Introduction

*L. monocytogenes* is a ubiquitous microorganism that can be present in ready-to-eat foods (RTE) since there is no bactericidal treatment during the manufacturing process [1] before consumption. Fresh cheese, as a RTE product are usually stored at refrigeration temperatures and since *L. monocytogenes* is a psychrotrophic bacteria can grow in this food [2,3]. According to [4], half of *L. monocytogenes* outbreaks reported in Europe have been associated to the consumption of dairy products. Previous research performed in Costa Rica, has confirmed *L. monocytogenes* contamination in artisanal fresh cheeses, raw milk and food contact surfaces within the processing environment; some of the strains isolated were found to have pathogenic potential (not published).

Dairy products are well-known for their high nutritional value, convenience and high diversity. In Costa Rica, the average consumption of dairy products is around 216 kg per capita per year [5]. An estimated 42% of the milk produced in Costa Rica is used for artisanal fresh cheese production [6]. “Turrialba cheese” is one type of “latino-type fresh cheese” and is considered one of the most representative dairy products in Costa Rica, being consumed by a large portion of the population and produced following a traditional fresh cheese process, using cow milk, culture and rennet. This cheese is mainly manufactured by artisans, making it common to use unpasteurized milk. Following operations include addition of ingredients, curding, draining, pressing and packaging. Salt is added after draining and currently on market is obtained with salt concentration between 1.5–2.0%. Reduction of sodium levels in its formulation could have a significant effect on consumers’ diet. The physico-chemical properties of this cheese include humidity below 55%, fat content above 18.5%, protein content above 14.5%, pH 6.2–6.5 [7], can favor the survival and growth of pathogenic and spoilage microorganisms. In addition, a large proportion of this type of cheese is elaborated under artisanal or semi-artisanal conditions where the lack of standard procedures and good manufacturing practices could enhance the presence of foodborne pathogens such as *Listeria monocytogenes* [8]. Artisanal cheeses in Latin America are frequently made with raw milk which represents an increased risk for the growth of pathogenic microorganisms. The effect of variation in “Turrialba cheese” formulation on the microbial safety of the final product has not been properly studied. A study involving 15 small artisanal producers from Costa Rica determined that 80% of them do not analyze the microbiological or physico-chemical properties of the milk used to elaborate their cheeses, and 53% of them have not yet established cleaning and disinfection procedures [9].

The risk of listeriosis associated with the consumption of “Latin-type fresh cheese” is related to the capacity of virulent strains to survive and proliferate on the final product. Even though *L. monocytogenes* growth on fresh cheeses has been demonstrated previously [10,11] to our knowledge, the capacity of *L. monocytogenes* to proliferate in “Turrialba cheese” has not been properly addressed. Predictive microbiology becomes a useful tool to characterize the response of microorganisms to environmental conditions [12]. Regarding salt concentration, there is an increasing demand to reduce the amount of sodium added to RTE foods [13,14]. However, this would compromise product safety since salt helps to curb *L. monocytogenes* growth, thus extending the shelf-life. This study aimed to develop a mathematical model to describe the effect of different salt levels on the growth of *L. monocytogenes* isolates from fresh cheeses.

## 2. Materials and Methods

Samples of “Turrialba cheese” (Latin-type fresh cheese) were obtained from different locations in Costa Rica to determine the levels of contamination with *L. monocytogenes*. Afterwards, isolates from cheese samples were used to perform challenge studies to analyze the effect of salt concentration (%) on the growth of *L. monocytogenes* during storage at refrigeration. Microbial counts (log CFU/g) were used to develop primary and secondary models to describe the growth potential of *L. monocytogenes* in “Turrialba cheese”. Finally, a model validation was also performed from the collection of data from previous publications, databases and predictive microbiology software tools (ComBase, Food Spoilage and Safety Predictor (FSSP) and Pathogen Modeling Program (PMP)).

### 2.1. Isolation of L. Monocytogenes from Cheese

#### 2.1.1. Collection of Samples

Sixty-four samples of fresh “Turrialba cheese” were collected from different retail stores located in the Central Valley of Costa Rica (more than 50% of the total Costa Rican population lives in this area) following a convenience non-probability sampling technique. Samples were collected preferentially from small shops in public markets in the cities of San José (*n* = 28), Heredia (*n* = 13), Cartago (*n* = 11) and Alajuela (*n* = 12). Samples were cut from a larger piece (unwrapped) at the moment of purchase. Each retailer was asked to provide fresher samples (less than six days old) as no specific label or information was available from any of the processors. Approximately, 200 g of each sample were collected inside a sterile sampling bag and transported at low temperature (<4 °C) to the laboratory for further processing. Samples were analyzed within 3 h of purchase.

#### 2.1.2. Isolation of *L. monocytogenes*

Minor modifications were made to the procedure detailed in the Bacteriological Analytical Manual [15] to isolate *L. monocytogenes* from fresh “Turrialba cheese. Briefly, 10 g of each sample was mixed and stomached with 90 mL of Buffered *Listeria* Enrichment Broth (BLEB; Oxoid, Basingstoke UK). This sample homogenate was incubated at 35 °C for 48 h. A total volume of 0.1 mL of the enriched broth was used to streak Modified Oxford Medium Agar plates (MOX; Oxoid, Basingstoke UK) both after 24 and 48 h of incubation. In addition, Fraser broth (Oxoid, Basingstoke UK) tubes were inoculated with 0.1 mL of the enriched BLEB (after 48 h of incubation) and incubated at 35 °C for 48 h, for additional enrichment. Additional MOX plates were streaked from the Fraser tubes that showed a dark color after the enrichment. After incubation, one typical *Listeria* spp. colony (gray color, central depression, black halo) from each positive sample was isolated on Tryptic Soy Agar (Oxoid, Basingstoke UK) plates for further identification.

#### 2.1.3. Identification of *L. monocytogenes*

Suspicious *Listeria* spp. colonies were typed initially using biochemical tests. Biochemical identification included the sugar fermentation test (xylose, rhamnose and mannose), the CAMP-test and the VITEK 2^®^ system (bioMérieux, Marcy-L’Etoile, France). Species identification was performed by proteomic analysis using Laser Assisted Matrix and Flight Time Desorption/Ionization Mass Spectrometry (MALDI-TOF/MS), using the MicroFlex LT system with MBT library DB-5989 (Bruker Daltonics, Bremen, Germany).

### 2.2. Challenge Test Study

#### 2.2.1. Manufacturing of Fresh “Turrialba Cheese”

One batch of Fresh “Turrialba cheese” formulated with different concentrations of salt (ranging from 0.5 to 5.2%) was manufactured at the pilot plant of Centro Nacional de Ciencia y Tecnología de Alimentos (CITA, University of Costa Rica). For this purpose, 60 L of fresh milk (pH values between 6.5–6.7) kept at a temperature of 5 °C was obtained from a local supplier. The milk was pasteurized at 63 °C for 30 min and cooled to 40 °C. This was followed by the addition of CaCl_2_ and the starter culture (strain STI-14; Christian Hansen, Denmark) to the milk once the temperature reached 37 °C; this mixture was left to rest for 25 min. The milk was then mixed with the rennet (CHYMAX-M; Christian Hansen, Denmark) and left to rest for an additional 25 min at 37 °C. The resulting curd was cut with a wireframe until proper granule measurement (1 × 1 cm) was obtained. The mixture was manually shaken for five minutes, and it was left to rest for ten minutes. The whey was separated from the curd manually and salt (%NaCl) was added at different levels according to the formulation (0.5; 0.8; 1.3; 2.0; 3.3; 5.2). The levels of salt were selected to simulate the average composition of “Turrialba cheese” and some extreme values (high and low). Finally, the cheese was placed inside cylindric molds and pressed using a pneumatic press (413,685 Pa) for 40 min. The final product was divided using sterile utensils into smaller 10 g portions that were air-packaged inside plastic bags and stored at 5 °C until analysis. The product obtained had the typical characteristics of fresh cheese: pH of 6.6, white color, moist appearance, soft milk odor, with small mechanical openings and low acidity perception on taste and odor. All the cheese batches were confirmed as negative for *L. monocytogenes*.

#### 2.2.2. Enumeration of *L. monocytogenes*

Different samples (10 g) of each cheese formulation were inoculated with a small population (3.0 log CFU/g) of a cocktail of five *L. monocytogenes* strains (M15, M21, M30, M31, M52) previously isolated as described in (Section 2.1.2). Samples were air-sealed inside plastic bags and stored at 4 °C. At different time intervals (1, 4, 8, 11, 15 and 18 days). Samples two of each “Turrialba cheese” formulation were taken until at least two points of the stationary phase were reached. The shelf-life of Turrialba cheese is between 7–18 days depending on the quality of the milk. *L. monocytogenes* populations were enumerated on solid media by using Modified Oxford Agar plates (MOX). The plates were then incubated at 35 °C for 48 h, and typical Listeria colonies were counted. Experimental data (log CFU/g vs. time) were used to build growth curves for each of the salt concentrations (%) evaluated.

#### 2.2.3. Physico-Chemical Characterization of Fresh “Turrialba Cheese”

Samples of the manufactured cheese were stored at 5 °C for seven days. At days 3, 5 and 7 three replicates samples of each salt concentration (%) were removed from storage and analyzed. Values of pH (method 981.12 AOAC [16] and water activity (a_w_ Decagon Devices) were measured during the storage of the fresh cheeses.

### 2.3. Modelling

#### 2.3.1. Primary Growth Model

Growth data of *L. monocytogenes* population (log CFU/g) were used to fit the Baranyi and Roberts primary model as this model is widely used to describe bacterial growth under static temperature conditions [17]. Equations (1) and (2) were used to estimate the four parameters model:(1)$$y\left(t\right)=y_{0+}\mu_{max}A\left(t\right)-\ln\left(1+\frac{e^{\mu_{max}A\left(t\right)}-1}{e^{\left(y_{max}-y_{0}\right)}}\right)$$
(2)$$A\left(t\right)=t+\frac{1}{\mu_{max}}\;\ln\left(e^{-\mu_{max}\;t}+e^{-\mu_{max}\;\lambda }-e^{-\mu_{max}\;(t+\lambda )}\right)$$

Being y(t) bacterial concentration (log CFU/g) at time *t* (days); $y_{0}\;$ and $y_{max}\;$ represent the initial and final bacterial concentration, also referenced as log N_0_ and log N_f_, respectively (log CFU/g); $\mu_{max}$ is the maximum growth rate, defined as the increase of microbial population per time unit (log CFU/d) also named (MGR, rate, *μ_max_*); and λ is the lag time, defined as the adaptation period before growth (hours or days), is the time elapsed from the inoculation of the microorganisms in the medium until they begin to multiply. The maximum population density (MPD) is the maximum level that the microbial population can reach under the conditions to which it is subjected (log CFU/g).

The experimental data were fitted using the DMFit tool from the ComBase software [18]. Model adjustment was evaluated by the pseudo- R^2^ parameter [19] and RMSE (root mean squared error). The kinetic parameters provided by DMFit for the growth data were used for secondary modelling.

#### 2.3.2. Secondary Growth Model

Maximum growth rates of *L. monocytogenes* ($\mu_{max})$ obtained from the primary model at different salt concentration (%) were fitted with different mathematical models using MATLAB R2021a; first and second order regression analysis was performed to test the relationship between the rate parameter and salt concentration (%). The commands used for performing this analysis were: R = corrcoef (A); p1 = polyfit (x, y, 2); f1 = polyval (p1, x); res1 = polyval (p1, x) − y); plot (x, y); plot (x, resl). Goodness of fit was assessed through the estimation of the coefficient of determination (R^2^) and root mean squared error (RMSE). The models developed were further implemented into the microHibro software tool (www.microhibro.com, accessed on 5 May 2021) in order to be accessible to users [20].

### 2.4. Determination of Growth Potential (δ) of L. monocytogenes

The growth potential (*δ*, log CFU/g) of *L. monocytogenes* at each level of salt concentration (%NaCl) was determined by the difference between the counts at the end of the experiments (day 11 at 0.5, 0.8 and 1.3 %NaCl); (day 15 at 2 % NaCl); and (day 18 at 3.3 and 5.2 %NaCl) and at the beginning (time 0) [21]. According to the Technical Guidance document for conducting shelf-life studies on *L. monocytogenes* in ready-to-eat foods (RTE) [21], when challenge testing is used to estimate *δ*, values higher than 0.5 log CFU/g are attributed to foods supporting the growth of *L. monocytogenes*. Therefore, this criterion has been applied to evaluate the different salt concentrations tested.

### 2.5. Model Validation

A comparison between data observed in this study and data collected from different databases was performed to validate the secondary model. Data were collected from ComBase [18], FSSP [22] and PMP [23]. In addition, data were also collected from previous publications. Data from previous studies were selected with consideration to food product characteristics, variables evaluated according to the behavior of the microorganism under study, and the target microorganism.

Afterwards accuracy and bias factors (*A_f_* and *B_f_*) were calculated according to Equations (3) and (4), respectively [24].
(3)$$A_{f}=10^{\frac{\sum \left|\log\left(\mu_{max}{}_{predicted}/\mu_{max}{}_{observed}\right)\right|}{n}}$$
(4)$$B_{f}=10^{\frac{\sum \log\left(\mu_{max}{}_{predicted}/\mu_{max}{}_{observed}\right)}{n}}$$
where *n* is the number of values obtained experimentally; $\mu_{max}$_*observed*_ are the values from reported studies on cheese and predictive modelling programs; $\mu_{max}$_*predicted*_ are the values estimated by the model obtained in this study.

The *A_f_* indicates the spread of data observed in comparison with the predictions of the model, and *B_f_* is a measure of how much the model under/or overestimates the kinetic parameter observed experimentally. Ideally, *A_f_* = *B_f_* = 1, which would indicate perfect agreement between the parameter values observed and the ones predicted by the model obtained. Graphical validation was performed by plotting the data obtained from the literature versus the predicted values of the model developed in this study, to assess the overall reliability of the model.

### 2.6. Statistics

Data analysis was performed with Microsoft Excel (Microsoft Corporation^®^) as average values and standard deviation, and IBM SPSS v25 (USA) to estimate the significant differences between the tested variables: pH and water activity (a_w_) with respect to the salt concentration (%NaCl) and storage time (days). Significant differences were considered using a *p*-value < 0.05.

## 3. Results

### 3.1. Listeria Monocytogenes Isolation from Fresh “Turrialba Cheese”

Table 1 shows the results for L. monocytogenes isolation from fresh “Turrialba cheese”. All the samples were obtained from small retail establishments where the cheese was cut at the moment of purchase. Information about the specific brand, the formulation and the expiration date of the samples were not available. It is probable that most of the samples originated from Turrialba, Cartago province. A total of nine of the 64 samples (14.1%) tested positive. Most of the samples were obtained from San José province but the highest levels of prevalence were observed in the case of Cartago (4 out of 11 positive samples). All samples obtained from Alajuela (No. samples 12) were negative for *L. monocytogenes*. Other Listeria spp. isolates were observed but their biochemical profile did not correspond to *L. monocytogenes*.

### 3.2. Challenge Test

The growth curves for L. monocytogenes in fresh cheese formulated with different concentrations of salt (%NaCl) are shown in Figure 1. It was observed that the microorganism was able to grow under all test conditions. It was expected that a characteristic lag phase of the growth curve would be identified, although no visible lag phase was recorded in any of the samples, and in all cases complete growth curves were observed. A lack of lag phase was obtained because the strains were adapted to the environment, the food product components and the salt concentrations used to make fresh “Turrialba cheese”. *L. monocytogenes* needed shorter times to attain stationary phase in the case of formulations containing lower salt concentrations (0.5, 0.8 and 1.3 % NaCl). For the formulation containing the lowest salt concentration (0.5%), the stationary phase was observed by day 5, and it appears that the results of day 8 are pulling down this phase estimate compared with the other salt concentrations, although the differences are less than 1 log. In the case of cheese containing 0.8% of NaCl, stationary phase was noticeable after ~7 days. On the other hand, L. monocytogenes stationary phase was reached after ~12 days on cheese samples containing a higher percentage of NaCl concentrations (3.3 and 5.2%). Results revealed that at a higher salt concentration (%), it took longer to reach the stationary phase.

Interestingly, the highest final L. monocytogenes populations were reported for some of the samples containing higher NaCl concentrations. For example, final L. monocytogenes populations of 9.37 ± 0.08 log CFU/g were observed in cheese formulated with 3.3% of salt whereas the maximum population in cheese containing 0.5% of NaCl was 8.18 ± 0.07 log CFU/g. However, cheese containing 5.2% of NaCl reported the lowest maximum population of L. monocytogenes (7.79 ± 0.16 log CFU/g) by the end of the sampling period. The total time needed to build complete growth curves was within the normal shelf-life reported on fresh cheese (15–18 days at 4 °C) in all the formulations.

#### Characterization of Fresh “Turrialba Cheese”

Table 2 shows the pH and a_w_ levels of fresh “Turrialba cheese” during storage at 4 °C. It was observed that levels of pH were not affected by the concentration of salt (*p* > 0.05) and the values decreased significantly (*p* < 0.0001) throughout storage. On the other hand, lower a_w_ values were reported for cheese samples containing higher NaCl levels as expected, and the lowest value was recorded for one of the samples containing 5.2% of salt (0.937); its levels remained constant for each formulation during storage (*p* > 0.05).

### 3.3. Modelling

#### 3.3.1. Primary Growth Model

The observed data of the growth of L. monocytogenes in cheeses were fitted to primary models using DMFIT from [18] as can be observed in Figure 1. The best fit was obtained with the Barany and Robers (Equations (1) and (2)) without lagtime. Table 3 shows the kinetic parameters obtained from primary models of L. monocytogenes growth in fresh “Turrialba cheese” with initial population level (*y*_0 obs_) of 1.93 log CFU/g. The R^2^ and Standard Error of fit (SE) of models ranged from 0.964–0.993, and 0.197 to 0.443, respectively.

An inverse relationship was observed between *µ_max_* parameter and salt concentration (%). (Table 3). Even though a lower value in those samples with highest salt concentration (5.2%) was observed, the maximum population density or final value does not seem to have a direct correlation with salt concentration.

The growth potential (*δ*_obs_) values were in the range of 5.86 to 7.45 log CFU/g, close to model prediction values that were from 5.69 to 7.25 log CFU/g. It is also noteworthy that the model predictions are generally more conservative, i.e., predictions are usually lower than real data (Table 3).

#### 3.3.2. Secondary Growth Model

Kinetic parameter growth rates obtained from the primary model at different % NaCl conditions were fitted (Figure 2) using a polynomial regression model in MATLAB R2021a. A second order polynomial equation was obtained as best fit (Equation (5)). An inverse relationship (*p* < 0.05) between rate parameter and salt concentration (%) was also reported. Correlation coefficients (R^2^) of 0.9391 was obtained. The results confirmed that the physical variable of salt concentration (%) has an effect on *L. monocytogenes* growth versus time on fresh “Turrialba cheese”.
(5)$$\mu_{max}=1.403-0.351*\left(NaCl\right)+0.035*{\left(NaCl\right)}^{2}$$

#### 3.3.3. Model Validation

The variable evaluated for validation was *µ_max_* of *L. monocytogenes* in cheeses with similar characteristics of this study (4 °C, pH = 6.5–6.7); and they were obtained from scientific publications and predictive programs software. If growth rates were not reported in the publications, they were estimated. If the conversion of a_w_ to salt % was considered necessary, it was calculated with the equation of [25].

The validation indices *A_f_* and *B_f_* (Table 4) was close to value 1 which indicates unbiased and accurate predictions [24]. For a pathogen such as L. monocytogenes, a high performance model is when *B_f_* is in a range of 0.95 to 1.11. An acceptable model performance is when *B_f_* is in a range of 1.11 to 1.43 or of 0.87 to 0.95 [24,26]. *B_f_* was slightly higher than value 1, meaning that estimations were fail-safe. According to the *B_f_* obtained (Table 4), the model developed in the current study was considered acceptable.

Results from the model validation performed with data from previous studies on cheese were considered acceptable (Table 4). In addition, the model validation performed from data collected using different software tools (Combase, FSSP and PMP) was classified as high-performance model validation. In Figure 3, it can be seen that most of the data were above the equivalence line.

In our case, *A_f_* in cheeses and in the modelling program is 1.43 and 2.73, respectively (Table 4). [24] stated that *A_f_* could increase by 0.10–0.15 for each variable included in the model, so we should expect values for *A_f_* of between 1.10 and 1.15 in a model incorporating the effects of salt on growth response.

The model has been introduced into microHibro “www.microhibro.com” (accessed on 5 May 2021) [20], an easy-to-use online application so that all agents, manufacturers or health authorities, can have easy access to the predictive models.

## 4. Discussion

### 4.1. Isolation of Listeria Monocytogenes from Fresh Cheese

RTE dairy products are a common vehicle for *L. monocytogenes* infection in humans [34]. Contamination of dairy products such as cheese normally occurs after processing, when some procedures such as cutting, and packaging are applied [35]. In the case of fresh cheese, contamination with *L. monocytogenes* could come from raw materials or food contact surfaces both at the industry and retail levels [36]. Latin-style fresh cheeses in particular are classified as high-risk foods for *L. monocytogenes* [37]. In fact, a study performed for the period 1998–2011 found 90 outbreaks linked to cheese, of which six were related to pasteurized fresh cheese [38].

In this study, the contamination of fresh cheese of Costa Rican origin (“Turrialba” style) was determined and it was confirmed that some products from retail outlets could harbor *L. monocytogenes*. A previous study performed in Costa Rica, confirmed the contamination of Turrialba fresh cheese although at a higher rate (24.5%) than in this study (14.1%) [39]. The study by [39] used a larger number of samples from different locations. It also reported the presence of other *Listeria* species, confirming that this food matrix is highly susceptible to microbial contamination. Turrialba cheese shares some of the common characteristics of other “Latin-type” fresh chesses in terms of no ripening, high levels of nutrients, high humidity (55–58%), high pH levels (5.0–6.3) and low salt content (1–3%) [36]. These conditions promote the survival and growth of *L. monocytogenes* during normal storage of the product at 4 °C [40].

As all the samples from our study were obtained from public stores and it could be assumed that contamination of fresh cheese is related to sanitary and handling practices performed at retail level [41]. However, a significant proportion of these Turrialba cheeses are fabricated under artisanal or semi-artisanal conditions where contamination from the processing environment may occur [42]. Furthermore, there may be significant variations in formulations (particularly the salt concentration), quality of raw ingredients and protocols for storage and distribution that may influence the final microbial safety and quality of the product.

### 4.2. Challenge Test in Fresh “Turrialba Cheese”

Given that fresh “Turrialba cheese” contamination with *L. monocytogenes* could occur, the study of pathogen behavior during storage is relevant in terms of risk assessment. Replication of *L. monocytogenes* in “Turrialba cheese” is expected given the capacity of *L. monocytogenes* to replicate under a wide range of temperature (1–45 °C), pH (4.0–9.5), a_w_ (>0.92) and salt (up to 10%) conditions [10]. According to [10], *L. monocytogenes* growth in Mexican-style cheese is possible at salt levels up to 9%, pH value above 6.0 and storage temperature of 10 °C. It is also noticeable that for all treatments, *L. monocytogenes* reached populations of >7.0 log CFU/g which is higher than the infectious dose reported for this pathogen [43]. In addition, active *L. monocytogenes* replication was observed in storage periods that are within the normal shelf-life span (15–18 days) reported for fresh cheese [44] that is kept at 4 °C. As “Turrialba cheese” is considered a RTE product it is possible that consumers will be exposed to infectious levels of *L. monocytogenes* as there may be no further cooking of the cheese.

Most of the available literature reports *L. monocytogenes* growth in cheese formulated with 2% salt and stored at 4 °C. For example, [45,46] report more that *L. monocytogenes* can reach populations of 7 log CFU/g in fresh cheeses stored for 11 days at 4 °C. Similar to this study [47] reported populations of more than 7 logs after 6 days of storage at 4 °C (Minas cheese with 2%) whereas [48] also obtained more than 6 logs after 7 days of storage at 7 °C (Queso fresco with 2% salt). The initial inoculation level (3.0 log CFU/g) used in the current study was high enough to cause an outbreak. As the strains were isolated from “Turrialba cheese” they were adapted to the food matrix. Additionally, the secondary model revealed that even if there was an inverse relationship between the maximum growth rate (log CFU/d) and the salt concentration (%), at the end of the evaluation period bacteria population presented high levels. Bacterial growth reported at the end of experimentation period was comparable to the levels reported in other studies [46,47].

Despite the fact that *L. monocytogenes* can grow easily in fresh cheese, few studies have attempted to understand the effect of salt concentration on the behavior of this pathogen. For this study, *L. monocytogenes* was able to grow under the highest level of salt (5.2%) but other studies demonstrated that salt levels as high as 9% can support *L. monocytogenes* growth depending on the pH level [49]. However, *L. monocytogenes* inhibition was observed in terms of growth rate values, meaning that the application of other conservation strategies [30], could have a significant synergistic inhibitory effect on *L. monocytogenes* growth on fresh cheese.

Values obtained for pH and a_w_ during different storage times and salt concentration are expected for fresh cheese [36,50]. Even though pH and a_w_ tend to decrease at higher salt concentration, fresh cheese is still considered a high-risk product due to the potential for microbial growth. Behavior of pH during storage time is also expected as biochemical changes are not common in this type of cheese due to its processing. However, pH decreases during storage could be linked to fermentation of residual lactose by heat-stable indigenous bacteria of milk, by the culture added or by post-processing contaminating bacteria [44]. Water activity, on the other hand, also showed an expected behavior.

In addition, differences were found among studies. In the study developed by [51] that included the evaluation of the effect of pH. *L. monocytogenes* growth versus pH was reported on cheese surfaces between 8 and 18 days. In addition, they affirmed that the lowest *L. monocytogenes* growth was found on the surface of cheeses with the lowest pH. If pH increased *L. monocytogenes* growth also increased on cheese rind. For these last results, values of pH reported were close to the pH range of “Turrialba cheese”. In contrast, [52] affirmed that no significant difference was found in *L. monocytogenes* survival between the low and standard salt concentrations at low and high pH; they also affirmed that *L. monocytogenes* growth decreased gradually as pH increased from 5.3 to 6.9 on low-salt concentrations at all study temperatures. Even though pH values were constant along the experimental work, these last results disagree with this study since growth rate decreases as salt concentration (%) increases.

### 4.3. Modelling

Maximum growth rate (*µ_max_*, log CFU/d) kinetic parameters of primary models obtained by this study agree with other studies as reported by [29]. Similar to our case, the Baranyi and Roberts model was fitted to the isothermal growth of inoculated pasteurized milk cheeses at a storage temperature of 4 °C. The water activity and pH were 0.96 and 6.40, respectively. Physicochemical characteristics of the food matrix make the studies comparable.

As expected, the increasing salt concentration (%), led to a reduction of *µ_max_* (log CFU/d) since salt concentration (%) affects *L. monocytogenes* growth. In the study conducted by [32], growth prediction of native strains of *L. monocytogenes* in fresh ricotta reported a maximum rate of 0.816 (log CFU/d) and *y_max_* of 8.62 (log CFU/g) with the conditions shown in Table 4. These results agree with the observed maximum growth rate (*µ_max_*, log CFU/d) but not with *y_max_* values (Table 3) on fresh “Turrialba” cheese with 1.3 and 2.0% salt. However, [30] reported values of *y_max_* (9.62–9.43 log CFU/g) that were slightly higher than our fresh cheese (9.236 log CFU/g) at more than 3% of salt concentration. In contrast, with similar salt conditions [29] reported lower values of *y_max_* (5.48 log CFU/g). Lower maximum growth rates (log CFU/d) of *L. monocytogenes* were described by [33], where only one of the strains used was isolated from Gorgonzola cheese. The growth rates ranged from 0.168 to 0.610 (log CFU/d). Furthermore, the origin and the different ability of the strains to adapt to environmental stresses and cheese structure characteristics are among the possible reasons for such differences.

In the study of [32] the growth of listeria was evaluated in mozzarella, and they affirmed that lag phase was not significant. Furthermore, the strains were isolated from human, a fresh dairy product and a dairy associated outbreak. These results are comparable with the current study, as lag phase was not observed and growth rates values were close. As *L. monocytogenes* strains in this study were isolated from fresh cheese samples, they probably adjusted and survived much better to the physical and chemical conditions of the fresh “Turrialba” cheese. This may explain why a lag phase was not observed. This contrasts with the studies of [35] that described lag time of 189.07 h with inoculated strains from clinical samples.

Growth potential (*δ*) for *L. monocytogenes* in fresh cheese was higher in comparison to other studies with similar product type (Table 3). Lower values of near to 6.50 log CFU/g were reported by [28,30] in fresh cheese and could be explained in terms of the absence of a lag phase given that only *L. monocytogenes* isolates from “Turrialba cheese” were used; these strains were probably better adapted to grow under the conditions given by the product.

On the other hand, lower growth potential has been reported (4.83–4.80) in the case of other dairy products which had lower pH values (5.42–5.57), such as pasteurized milk cheese paneer [27] and artisanal Belgian cheese [53] due to high acidity levels as shown by [32], in Mozzarella cheese after addition of citric acid.

The importance of the adaptation of *L. monocytogenes* native strains used in this study is also reflected by the fact that those studies using strains from other sources, such as [29] with 3.5% salt and [33] with 3.85% salt, show a lower growth potential (2.68, 1.80 log CFU/g, respectively) compared to ours (5.86 log CFU/g to 5.2% NaCl). Even though [54] inoculated native strains in their cheeses and reported a growth potential (4.87 log CFU/g) closer to our study, this corresponded to cheeses with more than 8% salt. These data confirm the importance of considering the origin of the isolates when evaluating the growth potential of this pathogen.

Finally, it is also important to note the capacity of *L. monocytogenes* to compete with other bacterial species. In general terms, *L. monocytogenes* is not a good competitor, and its growth potential (*δ*) is lower even when compared with other *Listeria* species such as *L. innocua* [55]. It is possible that the higher *L. monocytogenes* growth potential observed when salt levels increased is related with the inhibition of accompanying microbiota of fresh cheese. Some data collected in our laboratory (not yet published) suggest that the normal microbiota of fresh cheese is around 7.0 log CFU/g, meaning that microbial inhibition given by higher salt levels may offer a competitive advantage for *L. monocytogenes*. This is crucial, as factors promoting higher growth potential for *L. monocytogenes* may increase the risk for consumers specially in the case of RTE products, stored at 4 °C. Further research is necessary to evaluate how handling practices and the application of post-processing interventions may affect growth potential for *L. monocytogenes* in fresh cheese, and its relationship with accompanying microbiota.

In other foodstuffs, it is considered susceptible to pathogen growth, when the *δ* is higher than 0.5 log CFU/g [56], therefore, its classification falls into the potential risk categories when assessing a long shelf-life. Considering, that in our laboratory the lowest growth potential recorded was (*δ* = 5.86) for the different concentrations of salt, it can be stated that “Turrialba cheese” presents a high-risk potential to the consumer throughout the useful life.

### 4.4. Model Validation

According to [57] classification, the model developed by [33] had as a result a ‘very good’ model performance. The model performance of the other studies listed in Table 4 (experiments from [32]), are classified as ‘acceptable performance’ of the model as *B_f_* < 0.87 or > 1.43. The lack of a ‘very good’ model performance adjustment may be due to differences between temperature storage conditions and pH values established in previous studies. These variables are different from the storage temperature (4 °C) and pH value (6.6) of fresh “Turrialba cheese”. In addition, the origin and adaptation of the strains affected the validation results (*B_f_*).

In the case of different cheese product characteristics from fresh “Turrialba cheese” [28,29,30], the *B_f_* is closer to the general mean of validation 1.35. All the estimates of the fresh “Turrialba cheese” model are conservative, because the strains were isolated from this type of cheese and a *B_f_* > 1 always. As a result, this model estimates growth for the worst-case scenario, these results are related to the fact that the strains are already adapted to the conditions of the environment, this justifies the deviation of 1.43 obtained for *A_f_*. The same situation is observed in other studies such as [58] (*A_f_* 1.13 to 1.59) and [26] (1.78 to 1.33) where all *A_f_* are >1.

Previous studies researched the effect of temperature (°C), pH, water activity and time on *L. monocytogenes* growth on different cheese types, such as fresh ricotta [32], cottage [59], gorgonzola [33], fresh cheese [45] and others. However, there has not been much research on fresh chilled cheeses having different salt concentrations (%) and pH values close to neutral. As seen in Table 4, although the use of predictive models is increasing, further studies are needed on this since the availability of validated models that describe the behavior in foods is essential to prevent outbreaks. In addition, the model supports different steps of the food chain and other scientists who try to describe the behavior of *L. monocytogenes.* The current study contributes to knowledge on the effect of salt concentration (%) on *L. monocytogenes* growth on a fresh cheese type which is highly consumed in Central and South America.

Finally, the scientific contribution of this study also consists in the addition of this data into the free and easy-to-use predictive microbiology software “microHibro”.

## 5. Conclusions

This research allowed to quantify the contamination level and kinetic behavior of autochthonous isolated *L. monocytogenes* strains in “Turrialba fresh cheese”. Even though the application of predictive models, as microHibro app., helps food operators to set microbial limits and to determine food shelf-life, the effect of salt concentration in fresh cheeses has not been largely studied yet, especially in artisanal elaborated cheeses. Under the studied conditions, it was shown that the decrease of salt led to higher growth rate of *L. monocytogenes* reaching levels >7.0 log CFU/g at 4 °C regardless of the concentration of salt (%). Moreover, additional antimicrobial interventions may be considered for fresh “Turrialba cheese” to inhibit the growth of this pathogen. It is recommended that intervention strategies are combined with the development of standardized procedures and recipes to stablish low (healthy) salt concentrations (%) and simultaneously to ensure food safety for artisanal and semi-artisanal fresh cheese producers in Costa Rica.

Further studies are needed to integrate these data, since validated models are essential to describe pathogen behavior and to prevent outbreaks. The heterogeneous microbial milk quality and the different applied elaboration methods will need an additional characterization of microbial inter batch variability and processing conditions to properly determine safety shelf-life of “Turrialba fresh cheese”.

## Figures and Tables

**Figure 1 foods-10-01722-f001:**
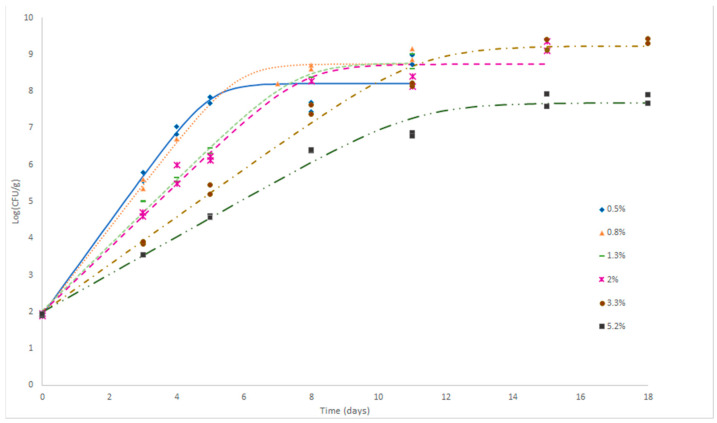
Growth curves of L. monocytogenes inoculated in fresh cheese with different %NaCl (0.5, 0.8, 1.3, 2.0, 3.3, 5.2) and stored at 4°C. The markers correspond to the observed data and the lines to the prediction curves
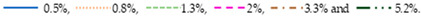
.

**Figure 2 foods-10-01722-f002:**
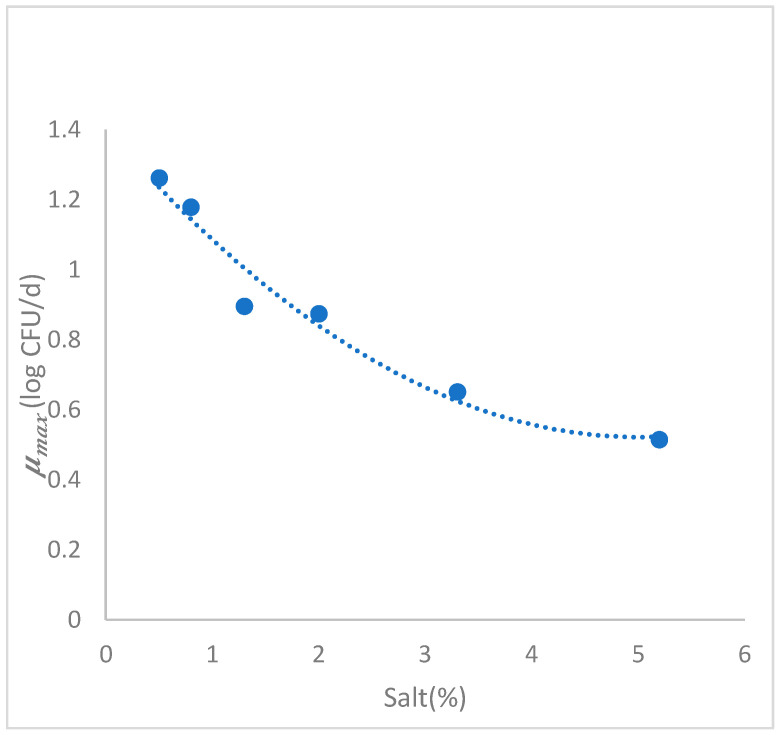
Graphical representation of the secondary model of maximum growth rate (log CFU/d) in function of the salt concentration (%) of Listeria monocytogenes in cheese.

**Figure 3 foods-10-01722-f003:**
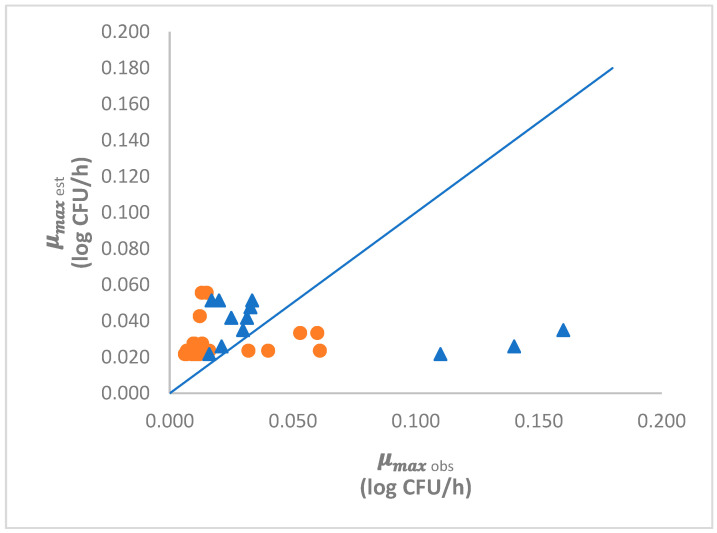
Graphical validation of the *µ_max_* observed (_obs_) vs estimated (_est_) by the secondary predictive model. (circle) Cheese studies; (triangle) Modelling Programs.

**Table 1 foods-10-01722-t001:** *L. monocytogenes* isolates obtained from fresh “Turrialba cheese” samples.

Province	Location *	No. of Samples	No. of Samples Positive for *L. monocytogenes*
San José	1	5	2
	2	7	0
	3	9	0
	4	2	0
	5	5	0
Cartago	6	2	0
	7	9	4
Heredia	8	5	0
	9	8	3
Alajuela	10	12	0
Total	-----	64	9

*. Shops locations: 1. Central market, 2. San José, 3. Desamparados, 4. Goicoechea, 5. Montes de Oca, 6. Cartago, 7. Cartago market, 8. Heredia, 9. Heredia market, 10. Alajuela market.

**Table 2 foods-10-01722-t002:** Characterization of fresh “Turrialba cheese” formulated with different NaCl levels during storage.

	3 Days			5 Days			7 Days		
%NaCl	pH	SD	a_w_	pH	SD	a_w_	pH	SD	a_w_
0.5	6.67	0.03	0.993	6.49	0.02	0.994	6.48	0.04	0.993
0.8	6.70	0.01	0.993	6.50	0.02	0.992	6.53	0.01	0.992
1.3	6.66	0.03	0.986	6.46	0.01	0.982	6.51	0.03	0.987
2.0	6.66	0.03	0.969	6.52	0.01	0.980	6.57	0.02	0.977
3.3	6.66	0.02	0.955	6.48	0.02	0.968	6.58	0.01	0.966
5.2	6.65	0.03	0.937	6.57	0.06	0.954	6.52	0.02	0.950

**Table 3 foods-10-01722-t003:** Values of maximum growth rate (*μ_max_*, log CFU/d), maximum population density (*y_max_*, log CFU/g), initial concentration (*y*_0,_ log CFU/g), estimated growth potential (*δ*_est_, log CFU/g), observed growth potential (*δ*_obs_, log CFU/g), obtained from the primary models of L. monocytogenes growth in fresh “Turrialba cheese”.

%NaCl	*μ_max_*	*y_max_* _obs_	*δ* _obs_	*y* _0 est_	*y_max_* _est_	*δ* _est_	R²	RMSE
0.5	1.262	8.851	6.93	1.93	8.21	6.28	0.964	0.443
0.8	1.178	9.000	7.08	1.94	8.74	6.80	0.992	0.254
1.3	0.895	8.805	6.88	2.04	8.78	6.74	0.993	0.197
2.0	0.874	9.230	7.31	2.00	8.74	6.74	0.978	0.368
3.3	0.651	9.370	7.45	1.99	9.24	7.25	0.989	0.293
5.2	0.514	7.785	5.86	2.00	7.69	5.69	0.986	0.257

**Table 4 foods-10-01722-t004:** Bias (*B_f_*); Accuracy (*A_f_*) factors and RMSE calculated for *µ_max_* values (log CFU/d) estimated by polynomial model of *L. monocytogenes* in fresh cheese.

References	Matrix (Broth/Type Cheese)	Factors	*B_f_*	*A_f_*	No.
[27]	Paneer	T(°C) = 4; pH = 5.42–5.57; a_w_ = 0.999; NaCl (%) = 0.19	1.82	1.82	2
[28]	“Queso Blanco” slices	T(°C) = 5; pH = 6.8; a_w_ = 0.971; NaCl (%) = 4.96	1.44	1.44	15
[29]	“Queso fresco”	T(°C) = 4; pH = 6.40; a_w_ = 0.96; NaCl (%) = 3	1.38	1.38	1
[30]	“Queso fresco”	T(°C) = 4; pH = 6.75; a_w_ = 0.983; NaCl (%) = 1.67	1.57	1.57	2
[31]	Cottage	T(°C) = 5; pH = 5.18; Salt% = 1.22; 718 ppm Lactic acid	1.72	1.72	1
[32]	Mozzarella with citric acid;	T(°C) = 4; pH = 5.49–6.43; a_w_ = 0.988; NaCl (%) = 2.18	0.80	1.25	2
[33]	Gorgonzola	T(°C) = 8; pH = 6.80; a_w_ = 0.978; NaCl (%) = 3.85	1.07	1.38	6
Total		RMSE = 0.0191	*B_f_* = 1.35	*A_f_* = 1.43	*n* = 29
**Modelling Program**	**Matrix (Broth/Type Cheese)**	**Factors**	***B_f_***	***A_f_***	**No.**
[18]	Broth media	T(°C) = 4; pH = 6.5–6.7; a_w_ = 0.969–0.997; NaCl (%) = 0.5–5.2	0.67	1.90	4
[22]	Cottage cheese	T(°C) = 5; pH = 5.5; a_w_ = 0.989; NaCl (%) = 2	1.15	1.15	4
[23]	Broth media	T(°C) = 4; pH = 6.6; a_w_ = 0.997–0.969; NaCl (%) = 0.5–5.1	1.24	1.24	4
Total		RMSE = 0.058	*B_f_* = 0.98	*A_f_* = 2.73	*n* = 12

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
