# Peer review of "Modelling the Effect of Salt Concentration on the Fate of Listeria monocytogenes Isolated from Costa Rican Fresh Cheeses"

_foods, 2021, doi:10.3390/foods10081722_

Round 1

Reviewer 1 Report

This study relates to the effect of salt concentration on the growth of Listeria monocytogenes in ‘Turrialba cheese’, a artisan Costa Rican cheese. While reporting of data relating to local cheese types is always welcome, the modelling approach use would now be considered routine and straight forward with no attempt to bring novelty to the modelling approach. Maybe some form of newer approach using a model that coupled a primary model for growth to secondary models for growth rate (C?F for example Juneja, V.K.; Cadavez, V.; Gonzales-Barron, U.; Mukhopadhyay, S. Modelling the effect of pH, sodium chloride and sodium pyrophosphate on the thermal resistance of escherichia coli O157:H7 in ground beef. Food Research International, 2015, 69, 289-304.)

A major reservation I have relates to both the conduct of the survey and the experimental approach to determine the growth curves. Table 1 sets out the number of samples obtained (64 in total). It would appear that only a limited number of retail outlets were actually visited (10 in total). It is not clear if these establishments were visited multiple times or whether all of the samples were taken at the same time from the one retail outlet. For example, for Establishment 3, nine samples were taken, were these taken all in one go?. If this is the case, it seriously reduces the value of the study as it is not clear what a sample is – is it different brands of the same cheese, is the cheese prepacked etc. Much more detail is required here.

For the growth studies, there is no clear statement relating to the level of replication of the work. Examining Figure 1, there appears to be two data points for each time point, but the closeness of the data points would indicate that these are ‘repeat’ measurements taken either as two separate 10 gram samples selected at each time point and tested separately or else two measurement carried out on the 10 g sample taken at the given time point. The closeness of the data points at time zero for all samples would indicate that there was no biological replicates undertaken in this study. i.e. whereby seprate batches of  fresh cheese are made up at different times and fresh Listeria inoculums were prepared from the selected strains for each new batch. A clear statement on the level of replication carried out is needed.

I also have reservations about the model validation section (3.3.4). This whole section is hopelessly vague and confusing. It is not clear what model is being validated. The only model actually presented in the paper is equation five which links salt content to max growth rate. So what is being validated?. Table 4 lists four factors from the studies collected, Temp, pH, aw and Salt %, but yet the experimental work was all done at one temperature so how can temperature suddenly be a factor? In addition, Figure 4 seems to be incorrect how can there be observed (circles) and estimated (triangles) when they are the x and y axes?

Minor comments

Line 65 replace elaboration process by manufacturing process

Line 70. Not a complete sentence, perhaps delete the while

Line 99 onwards. More description of how Turrialba cheese is made would be useful

Line 109 Isolates from cheese – replace with Cheese samples?

Line 118 64 samples

Is there any knowledge relating to the shelf life of these samples – was there any information about best before data and how much days were left?

120 ‘convenience non-probability sampling technique – is this an acceptable approach for a peer reviewed publication?

  1. Where the samples taken from a particular outlet all taken on the one day – were they from different brands, was the product wrapped/unwrapped – a bit more detail required

152 – some indication of how much cheese was made at a time. Was more than one batch of cheese at separate time periods made?

163 Use SI units for the pressure

166 Small mechanical eye – please explain or reword

168 How did you confirm the cheeses were negative for Listeria?

Section 2.2.2

Precise detail of level of replication required and if any repeat measurements were used

182 if only one sample was taken for ph and aw how was ANOVA carried out as reported in 3.2.1?

187 replace adjust with fit

189 Equation 1 and 2

Line 203 some lines of text appear to be missing

219 References 25 – 27 are missing in the reference section

Section 2.5

This needs a major rewrite – it needs to be explained precisely what model is being validated

240 this should be 2.6 not 2.5

More detail of what test statistics and on what data is required it appears to have been one way analysis on variance

264 There needs to be a discussion somewhere on why no lag period was observed – was this expected – what would combase predict for these temperatures, aw, ph and salt concentrations

Figure 1

There needs to be more discussion of what happened at the salt concentration of 0.5%. There appears to be two very low results at day 8 which is pulling down the stationary phase estimate. This needs to be highlighted This again points to the lack of proper replication.

278 The very small SD reported would point to no biological replicates

Table 2 – this data could be better represented as a figure

296 replace adjusted with fitted

298 – awkward referencing – please reword

300 why 1.925? the lowest value in table 3 is 1.93 and most values are closer to 2

305 Having plotted the final values versus salt concentration myself there appears to be no correlation between final value and salt concentration – please check your statistics

308 please include a definition of what model prediction values for growth potential is (presumably estimated final value minus estimated initial value)

Figure 2

Presumably the X axis is pH?

Section 3.3.4 Model validation

This is very muddled and impossible to follow. The definition of the data set collected (lines333) is vague and needs more precision. A clear definition of what model is actually being validated is necessary . How can temperature be included in the model as a factor when all of the experimental work was done at 4 Deg C?

Table 4 Values for actual and predicted growth rates should be included in the table. I would exclude Gorgonzola as it is a type of cheese very different to the Turrialba cheese’

Figure 4 seems to be incorrect how can there be different observed and estimated values?

Figure 5 This figure brings no utility to the paper. The choice of growth rate in the parameter section appear to be much smaller than what is reported in Table 3. There is no explanation as to what the differently coloured plot lines represent

4 Discussion

The discussion is far too long and at times repetitive. It is unfortunate that this section has no numbering and appears to be ‘inserted’ by one of the authors without proper integration with the rest of the paper.This section has to be drastically reduced to give a succinct comparison of the data with published data. Tables may help to summarise the model parameters with other published studies.

Second paragraph of Section 4.2 this paragraph is mostly a repeat of what is already in 4.1 please rewrite.

End of section 4.2 “However, L. monocytogenes inhibition was observed in terms of growth rate and maximum population reached” Please see my earlier comment - 305 Having plotted the final values versus salt concentration myself there appears to be no correlation between final value and salt concentration – please check your statistics – I don’t think this statement is supported by the evidence.

4.3 What is relevant here is a comparison of the growth rates, did the cited studies get growth rates comparable to what is reported in Table 2. Comparisons of R2 and so on is less relevant.

Author Response

We sincerely thank reviewers’ for their constructive comments, which in our opinion, have improved the quality and clarity of the revised manuscript. Please, find below a detailed response point by point to their queries.

Posada-Izquierdo et al. modeled the effect of salt concentration on the growth of pathogen isolates obtained from local cheese. Product from retail outlets in Costa Rica were analyzed for L. monocytogenes. These isolates were used to determine growth at 4°C for different salt concentration (0.5-5.2%). Kinetic curves were built and primary and secondary models developed. Finally, a validation study was performed using literature data.

Reviewer 1

Response to Reviewer 1 Comments

Comments and Suggestions for Authors

1.- This study relates to the effect of salt concentration on the growth of Listeria monocytogenes in ‘Turrialba cheese’, a artisan Costa Rican cheese. While reporting of data relating to local cheese types is always welcome, the modelling approach use would now be considered routine and straight forward with no attempt to bring novelty to the modelling approach. Maybe some form of newer approach using a model that coupled a primary model for growth to secondary models for growth rate (C?F for example Juneja, V.K.; Cadavez, V.; Gonzales-Barron, U.; Mukhopadhyay, S. Modelling the effect of pH, sodium chloride and sodium pyrophosphate on the thermal resistance of escherichia coli O157:H7 in ground beef. Food Research International, 2015, 69, 289-304.)

Response 1:

The authors thank the reviewer for this suggestion. In this case, a secondary model was built considering salt concentration as the only factor. As the one-step approach is surely a good alternative, in this paper the secondary model sufficiently explained the microbial response in this kind of cheese. As the final model is quite straightforward and easy to implement for both industrial and health authorities, we chose to keep the original modelling approach. Further studies considering additional factors may include the one-step modelling as the reviewer kindly suggested. The novelty of this work lies in the fact that the models are made with autochthonous strains isolated from this type of product, for which there is currently no scientific literature.

2.- A major reservation I have relates to both the conduct of the survey and the experimental approach to determine the growth curves. Table 1 sets out the number of samples obtained (64 in total). It would appear that only a limited number of retail outlets were actually visited (10 in total). It is not clear if these establishments were visited multiple times or whether all of the samples were taken at the same time from the one retail outlet. For example, for Establishment 3, nine samples were taken, were these taken all in one go?. If this is the case, it seriously reduces the value of the study as it is not clear what a sample is – is it different brands of the same cheese, is the cheese prepacked etc. Much more detail is required here.

Response 2: Just one sample were obtained per location. The legend of Table 1 was corrected to “Location” instead of “No of establishments”. In this sense, the number of samples per location corresponds with the number of stores that were visited for a total of 64. The samples are of different brands of the same “Turrialba cheese” (fresh cheese) in different prepacked conditions.

3.-For the growth studies, there is no clear statement relating to the level of replication of the work. Examining Figure 1, there appears to be two data points for each time point, but the closeness of the data points would indicate that these are ‘repeat’ measurements taken either as two separate 10 gram samples selected at each time point and tested separately or else two measurement carried out on the 10 g sample taken at the given time point. The closeness of the data points at time zero for all samples would indicate that there was no biological replicates undertaken in this study. i.e. whereby separate batches of  fresh cheese are made up at different times and fresh Listeria inoculums were prepared from the selected strains for each new batch. A clear statement on the level of replication carried out is needed.

Response 3:

The level of replication carried out was clarified in the text “At different time intervals (1, 4, 8, 11, 15 and 18 days) 2 samples of each condition of “Turrialba cheese” were taken until L. monocytogenes population was within at least two points of the stationary phase of growth curves”. As the reviewer points out, just one biological replicate has been performed in this study. The authors recognize that more replicates would yield to a more representative estimates of L. monocytogenes growth. However, the main objective was to primarily assess the growth ability of isolated L. monocytogenes strains in Turrialba cheese. Further studies will confirm the microbiological behaviour of the pathogen at different environmental conditions adding more biological replicates.

4.-I also have reservations about the model validation section (3.3.4). This whole section is hopelessly vague and confusing. It is not clear what model is being validated. The only model actually presented in the paper is equation five which links salt content to max growth rate. So what is being validated?. Table 4 lists four factors from the studies collected, Temp, pH, aw and Salt %, but yet the experimental work was all done at one temperature so how can temperature suddenly be a factor? In addition, Figure 4 seems to be incorrect how can there be observed (circles) and estimated (triangles) when they are the x and y axes?

Response 4:

Indeed, the validated secondary model is the only secondary model obtained. This model corresponds to equation 5, and it is being compared with some studies found with similar conditions of salt concentration and temperature in fresh cheese.

It has been modified in the text for a better understanding.

Minor comments

5.- Line 65 replace elaboration process by manufacturing process.

Response 5.-

It has been corrected in the text according to the reviewer's recommendation.

6.-Line 70. Not a complete sentence, perhaps delete the while

Response 6.-

It has been corrected in the text according to the reviewer's recommendation.

7.-Line 99 onwards. More description of how Turrialba cheese is made would be useful.

Response 7.-

A more detailed description of Turrialba cheese and how it is made was incorporated.

8.-Line 109 Isolates from cheese – replace with Cheese samples?

Response 8.

It has been corrected in the text according to the reviewer's recommendation.

9.-Line 118 64 samples

Response 9.

It has been corrected in the text according to the reviewer's recommendation.

10.-Is there any knowledge relating to the shelf life of these samples – was there any information about best before data and how much days were left?

Response 10.

Cheese samples from small shops are normally sold in less than 6 days. Still, the retailer was asked to provide fresher samples as no label or information was available from any of the original cheese processors. More clarification has been provided in the text.

11.-Line 120 ‘convenience non-probability sampling technique – is this an acceptable approach for a peer reviewed publication?

Response 11.

Central markets in major cities of Costa Rica are suitable places to obtain samples of artisanal Turrialba cheese. However, samples were purchased according with the observations made in situ where smaller outlets or those showing conditions that could promote contamination of cheese were selected (for example if the shop was not clean, there were refrigerators with open doors, or many different products were sold at the same time). As this study was aimed to provide a first approach on the levels of contamination of Turrialba cheese, we consider that the design of the sampling procedure was appropriate. Other articles in literature have used the same approach before (https://doi.org/10.1637/10873-052714-Reg)

12.-Line 122 Where the samples taken from a particular outlet all taken on the one day – were they from different brands, was the product wrapped/unwrapped – a bit more detail required

Response12

As clarified in Table 1 just one sample per outlet was obtained. However, several samples from a particular location were taken on the same day depending on the place. Some locations were visited in different days. No specific brand was identified as these cheeses are normally cut from a larger piece at the moment of purchase. More clarification was provided within the methods and materials section.

13.- Line152 – some indication of how much cheese was made at a time. Was more than one batch of cheese at separate time periods made?

Response 13

It was specified that one batch was elaborated at a time and 60 L of milk were used for each case.

14.- Line 163 Use SI units for the pressure

Response 14

It has been corrected in the text according to the reviewer's recommendation.

15.- Line 166 Small mechanical eye – please explain or reword

Response 15

It was reworded as “small mechanical openings”

16.- Line 168 How did you confirm the cheeses were negative for Listeria?

Response 16.

Each batch of cheese was confirmed negative for Listeria by using the same methodology explained in 2.1.2. More detail is provided within the text.

17.- Section 2.2.2 Precise detail of level of replication required and if any repeat measurements were used

Response 17. MR

It has been corrected in the text according to the reviewer's recommendation.

“Different samples (10 g) of each cheese formulation were inoculated with a small population (3.0 log CFU/g) of a cocktail of five L. monocytogenes strains (M15, M21, M30, M31, M52) previously isolated as described in (2.1.2). Samples were air-sealed inside plastic bags and stored at 4°C. At different time intervals (1, 4, 8, 11, 15 and 18 days) 2 samples of each condition of “Turrialba cheese” were taken until L. monocytogenes population was within at least two points of the stationary phase of growth curves. L. monocytogenes populations were enumerated on solid media by using Modified Oxford Agar plates (MOX). The plates were then incubated at 35°C for 48 h, and typical Listeria colonies were counted. Experimental data (log CFU/g vs. time) were used to build growth curves for each one of the salt concentrations (%) evaluated”.

18.- Line 182 if only one sample was taken for pH and aw how was ANOVA carried out as reported in 3.2.1?

Response 18.

All aw and pH data were pooled and differences as a function of storage time at each salt concentration were seen.

An ANOVA analysis with repeated measured was carried out as a function of salt concentration and storage time. For this case, three replicates were obtained per point for such measurements. Has been corrected in the text according to the reviewer's recommendation

19.- Line 187 replace adjust with fit

Response 19:

It has been corrected in the text according to the reviewer's recommendation.

20.- Line 189 Equation 1 and 2

Response 20

It has been corrected in the text according to the reviewer's recommendation.

21.- Line 203 some lines of text appear to be missing

Response 21

The missing text has been added.

22.- Line 219 References 25 – 27 are missing in the reference section

Response 22.

It has been corrected in the text according to the reviewer's recommendation and the reviewer Nº2 said: “Please restrict their number to around 50”.

“25.Beaufort, A. The determination of ready-to-eat foods into Listeria monocytogenes growth and no growth categories by challenge tests.  Food Cont. 2011, 22, 1498-1502. doi.org/10.1016/j.foodcont.2010.07.014.

  1. Ross, T. Indices for performance evaluation of predictive models in food microbiology. J. Appl. Bacteriol. 1996, 81, 501–508.
  2. Food Spoilage and Safety Predictor (FSSP). Available online: http://fssp.food.dtu.dk/ (Accessed on 05-05-2021).”

Section 2.5. 

23.-This needs a major rewrite – it needs to be explained precisely what model is being validated

Response 23.

It has been rewritten and corrected in the text according to the reviewer's recommendation. The first paragraph of section 3.3.4 has been moved to this section.

24.- Line 240 this should be 2.6 not 2.5

 Response 24

It has been corrected in the text according to the reviewer's recommendation.

25.- More detail of what test statistics and on what data is required it appears to have been one way analysis on variance

Response 25.

The authors thank the reviewer for this suggestion. This section was rephrased as: “Microbial counts against storage time were log-transformed and descriptive statistics were calculated in MS Excel (Microsoft Corporation). Further, one-way ANOVA analysis was carried out in IBM SPSS v25 (Chicago, Illinois, USA) to estimate the significant differences in microbial counts considered as independent variables salt concentration and storage time. Homogeneity of variances was assessed using the Levene and Tukey post-hoc tests were performed at 95% confidence level.”

26.-Line 264 There needs to be a discussion somewhere on why no lag period was observed – was this expected – what would combase predict for these temperatures, aw, ph and salt concentrations

Response 26.

This issue was addressed on the challenge test section in (section 3.2 and 4.2.)

Figure 1

27.-There needs to be more discussion of what happened at the salt concentration of 0.5%. There appears to be two very low results at day 8 which is pulling down the stationary phase estimate. This needs to be highlighted This again points to the lack of proper replication.

Response 27.

While it is true that the points of time 8 pull down the estimation of the stationary phase, this has been attributed to biological variability. However, the previous point and the back point are well defined. This clarification has been included in the text in paragraph 3.2

“For the formulation containing the lowest salt concentration (0.5%), stationary phase was observed by day 5, and it appears that the results of day 8 is pulling down this phase estimate compared with the others salt concentration, although these differences are less than 1 log”

28.- Line 278 The very small SD reported would point to no biological replicates

 Response 28.

2 samples of each condition of “Turrialba cheese” were taken. The authors recognize that more replicates would yield to a more representative estimates of L. monocytogenes growth. However, the main objective was to primarily assess the growth ability of isolated L. monocytogenes strains in Turrialba cheese. Further studies will confirm the microbiological behaviour of the pathogen at different environmental conditions adding more biological replicates.

29.-Table 2 – this data could be better represented as a figure

Response 29.

Because pH data does not present any type of trend, it makes no sense to represent it on a chart.

30.- Line 296 replace adjusted with fitted

Response 30.

It has been corrected in the text according to the reviewer's recommendation.

31.- Line 298 – awkward referencing – please reword

Response 31.

It has been corrected in the text according to the reviewer's recommendation.

32.- Line 300 why 1.925? the lowest value in table 3 is 1.93 and most values are closer to 2

Response 32.

It has been corrected in the text according to the reviewer's recommendation.

33.- Line 305 Having plotted the final values versus salt concentration myself there appears to be no correlation between final value and salt concentration – please check your statistics

Response 33.

The authors agree with this reviewer’s appreciation since it is true that no relationship was established between final concentration values and salt concentrations. Correlations were only referred to growth rates. This has been corrected in the revised version.

34.-Line 308 please include a definition of what model prediction values for growth potential is (presumably estimated final value minus estimated initial value)

Response 34.

This definition is found in section 2.4

Figure 2

35.-Presumably the X axis is pH?

Response 35.

It has been corrected in the figure 2 to the reviewer's recommendation.

Section 3.3.4 Model validation

36.-This is very muddled and impossible to follow. The definition of the data set collected (lines333) is vague and needs more precision. A clear definition of what model is actually being validated is necessary . How can temperature be included in the model as a factor when all of the experimental work was done  at 4 Deg C?

Response 36.

Temperature and other factors are not considered as a variable, only the concentration of salt is considered in our model. Studies with factors close to ours (4 °C, pH 6.5-6.7) have been sought in the literature and the factors mentioned in Table 4 correspond to each publication. The text has been revised according to the reviewer's recommendation.  

37.-Table 4 Values for actual and predicted growth rates should be included in the table. I would exclude Gorgonzola as it is a type of cheese very different to the Turrialba cheese’

Response 37

Although it is true that they are different cheeses, their physicochemical characteristics are very similar in terms of pH, salt and aw, and in fact the foundation of predictive microbiology is based on the study of the behavior of microorganisms as a function of intrinsic factors. . Furthermore, the Bf and Af of this cheese is very close to 1 (1.07 / 1.38), which indicates that it fits perfectly with our model.

Therefore, it was interesting to validate the model in this type of cheese different from ours.

38.-Figure 4 seems to be incorrect how can there be different observed and estimated values?

Response 38

It has been corrected in the figure 4 to the reviewer's recommendation.

39.-Figure 5 This figure brings no utility to the paper. The choice of growth rate in the parameter section appear to be much smaller than what is reported in Table 3. There is no explanation as to what the differently coloured plot lines represent

Response 39.

It has been deleted in the text according to the reviewer's recommendation.

4 Discussion

40.-The discussion is far too long and at times repetitive. It is unfortunate that this section has no numbering and appears to be ‘inserted’ by one of the authors without proper integration with the rest of the paper. This section has to be drastically reduced to give a succinct comparison of the data with published data. Tables may help to summarise the model parameters with other published studies.

Response 40

It has been rewritten and corrected in the text according to the reviewer's recommendation.

41.-Second paragraph of Section 4.2 this paragraph is mostly a repeat of what is already in 4.1 please rewrite.

Response 41

The authors thank the reviewer for this comment. However, we do not agree with this statement as second paragraph of section 4.2 refers to L. monocytogenes growth which is not mentioned in section 4.1. However, some changes were applied in second paragraph of section 4.2 to better clarify these ideas.

42.-End of section 4.2 “However, L. monocytogenes inhibition was observed in terms of growth rate and maximum population reached” Please see my earlier comment - 305 Having plotted the final values versus salt concentration myself there appears to be no correlation between final value and salt concentration – please check your statistics – I don’t think this statement is supported by the evidence.

Response 42.

It has been corrected in the text according to the reviewer's recommendation (see Response No. 33).

43.- Seccion 4.3 What is relevant here is a comparison of the growth rates, did the cited studies get growth rates comparable to what is reported in Table 2. Comparisons of R2 and so on is less relevant.

Response 43

This issue was addressed from the first to the third paragraphs in section 4.3 which corresponds to the modelling section on the discussion.

Reviewer 2 Report

Dear Madam

This manuscript is about the growth of Listeria monocytogenes in a Latin American artisanal cheese (Turrialba cheese) and the modelling of its growth on it. Some serious work has been performed in the construction of this model. Otherwise, the proof of Listeria occurrence in this cheese is somewhat loose. Considering these, the manuscript is of merit, with some polishing of the text needed by a person having English as a mother tongue.

Specific comments

The title does not fully correspond to the manuscript (… from artisanal cheeses), since only one type of cheese (Turrialba cheese) was examined.

P1 L23. Instead of pathogen isolates, simply isolates. No pathogenicity or even serotyping was performed in the isolates recovered.

P1 L30. “and” should not be in italics.

P2 L64-75. Start the Introduction part with this paragraph.

P2 L66-68. In addition, L. monocytogenes is halophilic, a characteristic that needs to be mentioned.

P2 L70-72. No reference, problematic syntax.

P2 L81-86. Please omit.

P2 L8-93. Summarize in max. two sentences and merge with the previous paragraph.

P2 L99-105. Please omit.

P2 L 107. Start with "Samples...".

P2 L 136-137. Was there any specific reason for picking only one colony? Usually up to five colonies are picked. This could lead to underestimation of the occurrence of Listeria monocytogenes.

P3 L141. Change “separated” to “typed” or something similar.

P5 L203. Probably there is something missing in here.

P5 L223-224. Literature is mentioned twice in the same sentence.

P7 Table 1. It is interesting to note that the positive samples are grouped in only three major establishments. Were there any similarities in these areas that could permit environmental contamination?

P9 L325. The dots should be removed.

P10 L328. “food” instead of “dood”.

P10 L346 – 350. These lines can be erased. Only the last one is of interest, which should be merged with the following paragraph.

P13 L 382. MicroHibro is easy to assess, therefore a snapshot of it is not needed.

Discussion. For some strange reason, Discussion has its own paging and there are no line numbers in this part.

Discussion (throughout). Please refrain from using expressions like "in a study..."of "In another study"  and use expressions like "In the study of Millet et al".

Discussion P1 L18. Listeria in italics.

Discussion P1 L 36. The prevalence of Listeria monocytogenes in Latin American cheeses is rather high. Perhaps the authors can add some information concerning foodborne disease in humans. Otherwise the prevalence in this type of cheese can be compared to other artisanal cheeses around the world with similar legislation norms, since e.g. in EU, there is a certain legislative provision for RTE products.

Discussion P5 "It should be noted that the isolates used for this study were not isolated from the same food product but clinical strains". Is there a reason to deduct that there are significant differences between clinical and food Listeria monocytogenes isolates?

The discussion of the results of [38] and [39] should not have formed a separate paragraph and of course a sentence should not start with the reference. Instead they should be shortened and form one paragraph with the following two paragraphs (up to "...the origin of isolates.").

References. There are 70 references in this article. Please restrict their number to around 50.

Round 2

Reviewer 1 Report

The authors have confirmed that only one biological replicate was carried out in this work. In my view, this is too little for a work worthy of publication. Consequently, I have no option but to recommend rejection.

Author Response

I have enclosed PDF file
